# Comparing the Perceptions of Gender Norms among Adolescents with Different Sibling Contexts in Shanghai, China

**DOI:** 10.3390/children9091281

**Published:** 2022-08-25

**Authors:** Chunyan Yu, Xiayun Zuo, Qiguo Lian, Xiangyang Zhong, Yuhang Fang, Chaohua Lou, Xiaowen Tu

**Affiliations:** 1NHC Key Laboratory of Reproduction Regulation, Shanghai Institute for Biomedical and Pharmaceutical Technologies, Shanghai 200237, China; 2Shanghai Jing’an Education College, Shanghai 200070, China

**Keywords:** gender norms, siblings, early adolescence, comprehensive sex education, China

## Abstract

Individuals’ gender development is influenced by the characteristics of personal and contextual environments. However, the role of sibling contexts in shaping gender norms has rarely been studied among Chinese youth at early adolescence as most of them were the only child. The aim of this paper is to compare perceived gender norms among adolescents aged 10–14 with different sibling configurations, to help inform and tailor guidance for sexual and reproductive health education in the future. We used the Global Early Adolescent Study baseline data collected from Shanghai, China. The sample for analysis was 1615 students. We used univariate analysis and multivariate ordinal logistic regression to compare perceived gender-stereotyped traits and gender role attitudes, stratified by age and sex. The results showed that sibling context was more influential for boys than girls at early adolescence in their gender socialization process. Among boys those who were with mixed-sex siblings scored higher on gender-stereotyped traits (*OR_only-child_*
_vs. *mixed-sex siblings*_ = 0.67, 95% CI: 0.48–0.94, *p* = 0.019; *OR_same-sex siblings_*
_vs. *mixed-sex siblings*_ = 0.59, 95% CI: 0.37–0.96, *p* = 0.033). Younger early adolescents aged 10–12 who were the only child or who had mixed-sex siblings perceived more traditional gender role attitudes than those living with same-sex siblings (*OR_only-child_*
_vs. *same-sex siblings*_ = 1.71, 95% CI: 1.06–2.75, *p* = 0.028; *OR_mixed-sex siblings_*
_vs. *same-sex siblings*_ = 1.74, 95% CI: 1.03–2.94, *p* = 0.037). Comprehensive sexuality education with gender and power components being well addressed, both in and out of the family, is needed to provide extra gender-inclusive and gender-egalitarian environments for youth.

## 1. Introduction

Gender is a complex, multidimensional concept [1]. Gender norms are societies’ spoken or unspoken rules about the acceptable behaviors of women and men, or girls and boys: how they should look, act, and even feel or think [2]. Socially constructed stereotyped gender traits often require males to be masculine and females to be feminine, while stereotyped gender roles govern what are valued and acceptable attributes for men and women. Such arrangements encourage the sexual dominance of males and subordination of females, causing sexual and reproductive health (SRH) issues including sexual risk-taking behaviors, sexually transmitted infections, unintended pregnancy, intimate partner violence, etc. [3,4]. Inequitable gender norms also contribute to many other adverse health risks beyond SRH issues, such as drinking, hypertension, self-rated health status and obesity [3,5,6].

### 1.1. Gender Norms and Early Adolescence

Worldwide, sexuality education has been promoted to provide formal and nonformal SRH information as part of promoting the well-being of adolescents since 1994 [7]. However, conventional sex education often addresses the importance of condoms/contraception but fails to emphasize gender and power [8]. While the endorsement of unequal gender norms and stereotypes is common in early adolescence [9], evidence has already shown that gender norms are at the center of social norms that influence adolescents’ sexual and reproductive health (SRH) and well-being outcomes [10].

Early adolescence is a critical developmental period for forming gender norms, as cognitive and physical changes and increased reinforcement of social expectations about gender during this period have significant implications for individual gender development [11,12]. Although this stage is a crucial window period, it was often ignored in the research region on gender inequalities and reproductive health [13].

### 1.2. Siblings’ Possible Impact on Early Adolescents’ Gender Development

The family context is essential to shape early adolescents’ perceived gender norms, providing the natural gender environments and directions that shape their gender concepts [14]. Siblings can play critical roles in gender socialization at early adolescence [15]. Most studies, however, have mainly focused on parental influences on adolescents. The sibling context, especially the only-child status of no siblings, has received very little attention with respect to the development of gender attitudes during early adolescence [16].

We speculate that siblings might influence perceived gender norms during early adolescence from limited evidence in at least two ways.

One is grounded in social learning theory. Siblings are an essential source of model or reinforcement of same-sex characteristics other than their same-sex parent [17]. Youth at early adolescence may observe and imitate gender-related information through their brothers or sisters, especially same-sex siblings. In other words, early adolescents with same-sex siblings will model gender-typical behaviors, leading to increased gender-stereotypical traits and conduct. However, mixed-sex sibling configuration in the family context may lead early adolescents to exhibit fewer gender-typical behaviors [18].

The other is from the ecological perspective, such that individual gender development could be influenced by the characteristics of their personal and contextual environments (e.g., sibling context) [19]. In contrast to social learning views, siblings may also serve as sources of social comparison [16]. For example, although boys and girls have been treated equally in many circumstances in recent years, parents with mixed-sex sibling configurations tend to use different parenting practices with their sons and daughters. Such practices may expose early adolescents with mixed-sex siblings to a more gender-typical environment than those with same-sex sibling configuration [20,21].

Previous research suggested that biological sex modulates the associations between siblings’ characteristics and perceived gender norms [22,23]. For instance, parents are often more concerned with the gender socialization of boys to show masculine behaviors than they are with girls [24]. Thus, even if early adolescents are in the same sibling context, perceived gender norms may differ for boys versus girls. Moreover, there are also gender differences in family norms and social expectations [23]. For example, in many developing countries, boys often have more freedom in their activities, while girls are usually under more control and expected to stay home to have more family protection than boys [23,25]. Therefore, compared to boys, girls may spend more time with parents and siblings—underscoring a different environment created by their biological sex.

### 1.3. Sibling Context and Gender Norms among Chinese Early Adolescents

Due to the One-Child Policy (OCP) enacted in 1979, China might be the country that has the largest proportion of only children worldwide [26]. The OCP implementation was somewhat influenced by region (e.g., ethnic minorities and rural families where the first child was a girl in some provinces were exempted from OCP), parental education level, family economic level and other factors [27]. The sixth national census data indicate that in 2015 there were 224 million only children. Of them, 87.3% were born in urban areas [28]. This group will still be significantly large even though the government has encouraged couples to have two or three children since the One-Child Policy was abolished in 2015 [29,30]. According to national family planning survey data, the proportion of only children dropped from 54.7% in 2006 to 37.0% in 2016 [31]. Thus, China will have increasing proportions of adolescents with same- or mixed-sex siblings in the coming decades.

Though Chinese culture is characterized by traditional Confucian beliefs, which have strict doctrines about unequal gender stratification and the distribution of power and resources between men and women [32], some studies suggest that gender norms are evolving nowadays [33]. For example, the percentage of girls engaged in higher education has increased rapidly, and more females have worked in more minor gender-typical careers since the late 1970s [33]. In addition, a cross-site study on gender norms has shown that early adolescents in Shanghai perceive more equal gender norms than those from Kinshasa, Indonesia and Cuenca [9].

### 1.4. The Current Study

Since there was a sizeable only-child population in China, the seemingly ambivalent gender expectations from traditional culture and modern demographic and economic development made it a unique sociocultural opportunity to explore Chinese early adolescents’ perceived gender norms. In addition, many researchers have stressed the importance of studying gender norms in diverse cultures, but gender-related research is primarily from North America or Western Europe [34]. Thus, we are particularly interested in the differences in perceived gender norms among Chinese only-child and non-only-child early adolescents.

Our focus in this study is to compare the perceived gender norms between only-child and non-only-child, specifically among those at early adolescence with different sibling contexts. The results could help guide sexuality education implementation and intervention for changing gender norms among Chinese early adolescents. We hypothesize that: (1) only-child early adolescents’ perceived gender norms may be more egalitarian/flexible compared to those only with same-sex siblings, but more traditional/stereotypical than those with mixed-sex siblings; (2) only-child early adolescents and those with same-sex siblings, grounded in view of social comparison, may have more egalitarian/flexible perceived gender norms than those with mixed-sex siblings.

## 2. Materials and Methods

### 2.1. Participants

Data used in this study were collected in Shanghai from November to December 2017, which is part of the baseline survey of the Global Early Adolescent Study (GEAS), a multi-center longitudinal research focused on gender norms among early adolescents living in disadvantaged urban sites [9,35]. The study subjects were adolescents aged 10–14 years and lived in a relatively poor community in Shanghai. We used a clustered sampling method. First, we selected three secondary public schools from the target district in Shanghai. Then, we invited all students of the 6–8th grades in these schools to join in the survey to achieve our sample size. We obtained informed consent from all study subjects and their parents before conducting the study. School teachers helped us organize the students to complete the Computer-Assisted Self-Interview survey using tablets in their classrooms. Participants filled out the questionnaire independently. Investigators, trained on survey procedures, tools and other related issues, were allowed to assist respondents only if there were understanding or technical problems. The Computer-Assisted Self-Interview approach via the tablets contributed to the program consistency checks and skip-patterns to minimize errors [32], and protected participants’ privacy and anonymity. A total of 1775 students completed the survey. We excluded 47 respondents who were missing more than 15% of the responses in the whole survey, as such might indicate poor data quality. Other criteria included: (1) respondents were 10–14 years; (2) there were nothing missing in the early adolescents’ sibling context; and (3) there were nothing missing in the outcome—in either one of the perceived gender norm subscales. The eligible final sample was 1615 students.

The questionnaire used in the study was developed by the GEAS global team [9], and included early adolescents’ perceived gender norm scales and related influencing factors such as sex, age, birthplace and family context. The GEAS instruments could be accessed at https://www.geastudy.org/download-the-measures (accessed on 22 July 2022).

### 2.2. Instruments

#### 2.2.1. Gender Norms Scales

The main outcomes of our interest are two gender norms subscales: Gender Stereotyped Traits (GST) and Gender Stereotyped Roles (GSR), which are composed of 11 items responded to with a 5-point Likert scale (1: Disagree a lot, 2: Disagree a little, 3: Neither agree nor disagree, 4: Agree a little, 5: Agree a lot). Each individual’s sub-scale score was computed as a mean score across items. The GST subscale has seven statements, and higher GST scores reflect more stereotypical attitudes related to masculinities and femininities. The GSR subscale consists of four items, and higher scores indicate that individuals hold a less egalitarian or more traditional attitude toward the division of labor between men and women. The statements of each subscale are listed in Table 1. The Cronbach’s α of each subscale are 0.70 and 0.74, respectively. To better capture the influence of sibling configuration, we split the continuous mean scores of each subscale into tertiles in the ordinal logistic regression.

#### 2.2.2. Sibling Context

Early adolescents’ sibling context was categorized as only-child or non-only-child. The only-child was defined as the only biological child of the parents in the family, while the non-only-child was further separated into a same-sex siblings configuration (if they only had same-sex siblings) and a mixed-sex siblings configuration (if they had at least one opposite-sex sibling).

#### 2.2.3. Stratified Variables and Controlling Variables

Personal characteristics: we included individual demographic variables such as sex (male versus female) and age group (10–12 years old versus 13–14 years old) [36]. We regarded participants’ age group and sex as stratified variables.

Family context: we included birthplace (Shanghai or other cities) [37] and parental marital status (married versus widowed/divorced/separated) in the family context which may have the potential to influence and shape perceived gender norms [38]. We then included the socioeconomic status of participants’ families, such as the mother’s work status (working currently or not), and the mother’s or father’s highest educational level (entered or completed university versus high school/trade school or technical school/secondary school/primary school), as they might have an influence on gender socialization [39].

### 2.3. Data Analysis

Stata/SE 15.1 was used for data analysis. Descriptive analysis was used to compare the demographic characteristics of the only-child and non-only-child. Means and standard deviations were adopted to describe each subgroup’s gender norms. Univariate analyses were used to examine the sex and age differences in subjects’ scores on two gender norms subscales among early adolescents of different sibling contexts. Finally, multivariate ordinal logistic regression, stratified by sex and age group, was performed to examine the difference in perceived gender norms among only-child early adolescents and those with mixed/same-sex sibling configuration.

### 2.4. Ethical Considerations

APA ethical standards were followed in the conduct of the study, including measures design, research approach, consent protocols and procedures to protect the confidentiality and rights of all participants. The study protocol was reviewed and approved by the local institutional review board. Informed consent was obtained from all subjects involved in the study and their legal guardians.

## 3. Results

Among the 1615 participants, 1044 were only-child early adolescents, and 571 were non-only-child (163 were of same-sex sibling configuration, while 408 were of mixed-sex sibling configuration). About 50.71% of the participants were 10–12 years old, and 50.15% were boys. Over half of the participants (54.34%) reported their parents’ highest educational level as having entered or completed college. A majority (89.26%) said that their mothers were working for pay currently, and 84.79% were born in Shanghai. Only a small proportion of the early adolescents (11.80%) reported their parents being widowed/divorced/separated (see Table 2).

In the univariate analysis among the non-stratified sample, we did not detect any significant differences in either the stereotyped gender traits or the stereotyped gender roles subscales among only-child and non-only-child early adolescents with same-sex siblings or mixed-sex siblings at the 0.05 level. When stratified by age group, among the age group 10–12 years old, both early adolescents of only-child and non-only child with same-sex siblings scored lower than their counterparts with mixed-sex siblings on GST (see Table 3). Among the 13–14 years old group, non-only-child early adolescents with same-sex siblings scored higher on the subscale measuring GSR than only-child early adolescents, indicating that only-child early adolescents were less traditional regarding the division of labor between men and women. When stratified by sex, we did not find any differences in GSR scores among early adolescents of the three different sibling contexts. However, we observed higher GST scores in non-only-child boys with mixed-sex siblings than only-child boys, suggesting more stereotyped views about masculinities and femininities being accepted by non-only-child boys with mixed-sex siblings. On the other hand, such associations were not observed among girls of different sibling contexts.

In the multivariate ordinal logistic regression stratified by age groups, we detected that boys held more stereotyped attitudes towards gender-related traits and roles than girls in both the older (13–14-year-old) and the younger (10–12-year-old) age groups (*p* < 0.001). Among the younger age group, in contrast to the univariate analysis result, we only found that early adolescents with same-sex siblings held less stereotyped attitudes towards gender traits than those with mixed-sex siblings (*OR_mixed-sex siblings_*
_vs. *same-sex siblings*_ = 1.75, *p* = 0.026). Similarly, younger early adolescents with same-sex siblings held less traditional and more egalitarian attitudes toward gender roles regarding the division of labor between men and women than their counterparts with no siblings or mixed-sex siblings (*OR_only-child_*
_vs. *same-sex siblings*_ = 1.71, *p* = 0.028; *OR_mixed-sex siblings_*
_vs. *same-sex siblings*_ = 1.74, *p* = 0.037). The association between sibling context and GST/GSR was not found among the older age group (see Table 4).

In the multivariate ordinal logistic regression stratified by sex, both only-child boys and boys with same-sex siblings perceived less stereotypical gender traits than boys with mixed-sex siblings (*OR_only-child_*
_vs. *mixed-sex siblings*_ = 0.67, *p* = 0.019; *OR_same-sex siblings_*
_vs. *mixed-sex siblings*_ = 0.59, *p* = 0.033). For girls, however, the association between only-child versus same/mixed-sex sibling configuration and GST scores was not significant. Moreover, among girls, if their mothers were working currently, such a family context would have marginal contributions (*OR* = 0.68, *p* = 0.076) to girls’ more egalitarian attitudes toward gender roles regarding the division of labor between men and women (see Table 5).

## 4. Discussion

In this article, we charted the perceived gender norms among early adolescents with different sibling contexts with respect to stereotyped gender traits and gender role attitudes, reflecting two facets of the gender concept [22,23,36]. To our knowledge, this was the first published study to evaluate the difference in perceived gender norms between only-child and non-only-child early adolescents in the Chinese context.

Bivariate results showed that non-only-child younger early adolescents with mixed-sex siblings and non-only-child boys had more stereotyped views than their only-child counterparts. However, all early adolescents had somewhat gendered stereotyped views about sex-typed personality qualities (mean scores of GST above 3—“Neither agree nor disagree”). Our findings suggested that younger early adolescents with mixed-sex siblings were associated with more stereotypical gender attitudes towards gender traits than either only-child younger early adolescents or early adolescents with same-sex siblings. This finding was consistent with the social comparison view [16]. Early studies had demonstrated that gender comparisons, which emphasize the different traits of males and females, were more common in families with mixed-sex siblings [16,21]. Gender-differentiated parenting is more common when there are opposite-sex siblings, which may provide a more stereotypical family background that results in a more stereotypical gender belief [40,41]. In contrast, only-child early adolescents and early adolescents with same-sex siblings may hold less gender-stereotyped attitudes due to the lack of gender comparison sources. Our multivariate findings only supported the statistical differences between early adolescents with mixed-sex siblings and same-sex siblings. It did not show significant variance between either only-child early adolescents and those with mixed-sex siblings or only-child early adolescents and those with same-sex siblings. The reason may come from extrafamilial agents. One previous study in China underscored that only-child adolescents often have conventional peers even though they do not have siblings [42]. However, only-child adolescents may have more opportunities to associate with same-sex peers as stereotypical gender norms make a strict gender boundary in friendships [43]. Besides, parents often underscore the restriction of interacting with the opposite sex as their only child gets older [25]. That might explain why only-child early adolescents endorse the moderated gender-stereotyped traits. Most only-child early adolescents may observe/imitate similar gender-stereotypical behaviors with early adolescents who have same-sex siblings, while some may have opportunities to associate with friends of the opposite sex.

Regardless of the gender stereotypical sibling environment or observed gender-differentiated parenting, we did not find significant differences in early adolescent girls’ perceived gender-stereotyped traits among non-only-child with mixed-sex siblings, non-only-child with same-sex siblings or only-child adolescents. This might be because of the dominant place of hegemonic masculinity: boys were taught to endorse gender norms about physical toughness (e.g., more tolerable, fighting with others) and emotional stoicism (e.g., don’t behave like a girl) than girls [34,44]; femininity norms, in contrast, are less restrictive than masculinity norms in early adolescence [44], suggesting that sex is indeed an essential moderator of the associations between sibling background and perceived gender norms.

Comparing gender role attitudes towards the division of labor between men and women, we found that both only-child early adolescents’ and non-only-child early adolescents’ gender roles attitudes were less traditional due to evolving gender norms. Among early adolescents aged 13–14, only-child early adolescents held relatively less traditional gender roles attitudes than non-only-child early adolescents with same-sex siblings, although the mean scores for all three groups were below three (“Neither agree nor disagree”). However, when adjusting for other covariates in multivariate analysis, such differences were no longer significant among the 13–14-year-olds group. Interestingly, in the age group 10–12, early adolescents with same-sex siblings were less traditional than the other two types of sibling contexts regarding the division of labor. Younger early adolescents would learn from gendered family environments (e.g., gender-differentiated parenting, the gender-stereotyped environment created by different sibling contexts or the role models built by their parents) if they have no other source of sex comparisons. Early adolescents with same-sex siblings had fewer opportunities to be exposed to gendered environments than those with the other two types of sibling context, which might explain why they were less traditional. The additional sources might make a difference when early adolescents get older, while the differences in gendered family environments were probably negligible. Child and adolescents’ gender development, as ecological theory suggested, relied on the combination of multi-level factors [45]. Early research reflected that children with different sibling contexts would look for extrafamilial role models [22]. Additional sources of gender socialization (e.g., media, school) could also influence adolescents’ gender development [46], leading to only-child early adolescents, those with same-sex siblings and those with mixed-sex siblings being exposed to similarly gendered environments.

Being a girl was the only statistically significant protective factor in shaping an egalitarian attitude towards the division of labor between men and women among 10–14-year-old early adolescents, possibly reflecting that gender role attitudes in China are evolving. On the other hand, it might result from economic development and the need for women “to hold up half the sky”. Such an assumption was also echoed in the marginal protective effect of a mother’s currently working status (vs. not working) on less traditional gender role attitudes among girls. Due to the implementation of the One-Child Policy since the late 1970s, females in modern society are expected to be well educated and economically independent [25]. However, inequitable gender norms regarding “men are bread-earners and women are housekeepers” were deeply rooted in ancient China. Furthermore, we found that boys held more traditional attitudes regarding the division of labor between men and women than girls, indicating that the traditional gender norms in China were still in play [26]. It has also been revealed by evidence from other countries that boys are less likely to challenge the traditional gender norms as these norms often privilege boys [22].

## 5. Limitations

Since gender norms may vary across cultural contexts, our study provides a more comprehensive understanding of those gender norms subscales among early adolescents in the Chinese context [36]. However, there were still some limitations of this study that need to be noted. First, although we charted the perceived gender norms among only-child early adolescents of different age groups, these changes may be cohort differences due to our cross-sectional design instead of changing along the period. Second, our study only considers the sibling context in the way of being same-sex or not. Further studies should explore the association between sex and sibling contexts, not only the effects of sibling sex but also the birth order and their combination [34,47]. Third, our hypothesis related to differences in perceived gender norms between only-child and non-only-child early adolescents was based on the gender comparison view and social learning theory within the family context. However, the result that there are similarities in gender stereotypes between only-child early adolescents and those with same-sex siblings could not be explained by the existing selected variables, suggesting that there may be additional influencing sources or other mechanisms that need to be considered. Fourth, although the Computer-Assisted Self-Interview approach was a helpful collecting tool that could improve the answers’ accuracy, our self-interviewing data may still be influenced by recall bias. Lastly, as the number of observations on factors like mother’s work status, place of birth and parental marital status is small and we attempt to use logistic regression adjusting for several factors, it can fail to produce sensible results. Moreover, given the nature of the cross-sectional data, the results of our study could not guarantee the causal interpretation of the findings nor their generalizability beyond the urban poor early adolescents in this geographical area.

## 6. Conclusions and Implications

Our finding that only-child early adolescents in the younger age group scored in the middle level compared to early adolescents of two different sibling contexts regarding the two gender norm subscales suggested that gender norms are malleable. Early adolescents are not passive receivers that absorb all gendered information from others. Instead, they also process the messages inconsistent with prior self-concept, which are often ignored or transferred [18]. Emphasizing gender and power is the key to reducing sexually transmitted infections and unintended pregnancy [48]. Therefore, the gender development of children, especially those with same-sex siblings, must be noticed. The World Health Organization pointed out that sexual health is critically influenced by gender norms, roles, expectations and power dynamics [49]. Yet there is mounting evidence that Comprehensive Sexuality Education (CSE), with the component addressing issues of gender and power, is particularly effective [50,51,52]. UNESCO technical guidance regarding CSE could be delivered in formal and non-formal settings, and it is incremental, as well as age and developmentally appropriate. CSE contributes to gender equality by building awareness of the centrality and diversity of gender in people’s lives; by examining gender norms shaped by cultural, social and biological differences and similarities; and by encouraging the creation of respectful and equitable relationships based on empathy and understanding [53]. Such guidance could be used by parents, schoolteachers, health workers and gender professionals to create a more gender-inclusive context for children and adolescents. For parents, sometimes being supportive doesn’t mean doing anything more than simply providing a loving and affirmative place for children to express themselves. For schoolteachers, building a supportive relationship with respect and equal treatment for all students, and teaching about equitable gender attitudes using lessons, materials and vignette scenarios, would help the understanding of gender equity. Health workers and gender professionals could offer a professional response to the needs of both children and their families, and of other care providers, to accompany their growth and comprehensive development, by respecting their individuality and advocating for the full exercise of children’s and adolescents’ gender and power equality. At the societal level, promoting laws and policies to assure gender equality, as well as organizing media campaigns and large-scale communication programs, can contribute to raising awareness [54]. In all, inequitable gender norms need to be challenged from the macro-level (e.g., promoting laws and policies) to the micro-level (e.g., education both in school and at home) [55].

## Figures and Tables

**Table 1 children-09-01281-t001:** The items of early adolescents’ gender norms scale developed by GEAS research group.

Sub-Scales	Items
Gender Stereotypial Traits (GST)	(1) Boys should be raised tough so they can overcome any difficulty in life
	(2) Girls should avoid raising their voice to be lady like
	(3) Boys should always defend themselves even if it means fighting
	(4) Girls are expected to be humble
	(5) Girls need their parents’ protection more than boys
	(6) Boys who behave like girls are considered weak
	(7) It’s important for boys to show they are tough even if they are nervous inside
Gender Stereotypical Roles (GSR)	(1) A woman’s role is taking care of her home and family
	(2) A man should have the final word about decisions in the home
	(3) A woman should obey her husband in all matters
	(4) Men should be the ones who bring money home for the family, not women

Note: Reproduced with permission from [9].

**Table 2 children-09-01281-t002:** Distribution of demographic characteristics among only-child and non-only-child early adolescents (*n* (%)).

Variables	Groups	Total (*n* = 1615)	Only-Child (*n* = 1044)	Non-Only-Child		
				Total (*n* = 571)	With Same-Sex Siblings (*n* = 163)	With Mixed-Sex Siblings (*n* = 408)
Age	10–12	819 (50.71)	524 (50.19)	295 (51.66)	92 (56.44)	203 (49.75)
	13–14	796 (49.29)	520 (49.81)	276 (48.34)	71 (43.56)	205 (50.25)
Sex	Boy	810 (50.15)	508 (48.66)	302 (52.89)	93 (57.06)	209 (51.23)
	Girl	805 (49.85)	536 (51.34)	269 (47.11)	70 (42.94)	199 (48.77)
Parents’ highest educational level **^,#^	High school or below	683 (45.66)	417 (42.95)	266 (50.67)	73 (48.03)	193 (51.74)
	Enter or complete college	813 (54.34)	554 (57.05)	259 (49.33)	79 (51.97)	180 (48.26)
Mother’s working status *^,#^	Working for pay	1405 (89.26)	930 (90.64)	475 (86.68)	144 (90.00)	331 (85.31)
	Not working	169 (10.74)	96 (9.36)	73 (13.32)	16 (10.00)	57 (14.69)
Born in Shanghai ***^,###^	Yes	1355 (84.79)	940 (90.47)	415 (74.24)	119 (75.80)	296 (73.63)
	No	243 (15.21)	99 (9.53)	144 (25.76)	38 (24.20)	106 (26.37)
Parental marital status ***^,###^	Married	1413 (88.20)	946 (91.22)	467 (82.65)	134 (82.72)	333 (82.63)
	Widowed/divorced/separated	189 (11.80)	91 (8.78)	98 (17.35)	28 (17.28)	70 (17.37)

* *p* < 0.05; ** *p* < 0.01; *** *p* < 0.001. Comparison between only child and non-only child. ^#^
*p* < 0.05; ^###^
*p* < 0.001. Comparison among only child, non-only child with same-sex siblings, and non-only child with mixed-sex siblings.

**Table 3 children-09-01281-t003:** Mean scores ($\bar{x}$ (*sd*)) for gender norms scales of participants by different sibling contexts, stratified by sex and age group.

	*n*	Only Child (*n* = 1044)	With Same-Sex Siblings (*n* = 163)	With Mixed-Sex Siblings (*n* = 408)	*F*	*p*
GST						
Total (($\bar{x}$ (*sd*))	1606	3.30 (0.71)	3.33 (0.75)	3.37 (0.79)	1.35	0.259
Stratified by age group						
10–12 years old	814	3.30 (0.75) ^$^	3.23 (0.79) ^#^	3.45 (0.79) ^$,#^	3.37	0.034
13–14 years old	792	3.30 (0.66)	3.45 (0.69)	3.30 (0.80)	0.85	0.501
Stratified by sex						
Male	804	3.41 (0.71) ^$^	3.39 (0.72)	3.56 (0.78) ^$^	3.20	0.041
Female	802	3.20 (0.38)	3.25 (0.79)	3.18 (0.77)	0.22	0.799
GSR						
Total (($\bar{x}$ (*sd*))	1593	2.60 (0.94)	2.65 (0.98)	2.67 (0.95)	0.84	0.432
Stratified by age group						
10–12 years old	809	2.58 (0.96)	2.46 (0.89)	2.63 (0.94)	1.05	0.349
13–14 years old	784	2.61 (0.93) ^&^	2.90 (1.04) ^&^	2.70 (0.96)	3.04	0.048
Stratified by sex						
Male	795	3.02 (0.88)	2.94 (0.95)	3.08 (0.95)	0.81	0.445
Female	798	2.20 (0.82)	2.25 (0.88)	2.24 (0.75)	0.28	0.757

Note: ^$^: post hoc Bonferroni test between “only-child” group and “with mixed-sex sibling” group, *p* < 0.10. ^#^: post hoc Bonferroni test between “with same-sex sibling” group and “with mixed-sex sibling” group, *p* < 0.10. ^&^: post hoc Bonferroni test between “only-child” group and “with same-sex sibling” group, *p* < 0.10.

**Table 4 children-09-01281-t004:** Ordinal logistic regression testing sibling context on two gender norm scales’ scores, stratified by age group.

	10–12 Years Old						13–14 Years Old					
	GST			GSR			GST			GSR		
	OR (95% CI)	*z*	*p*	OR (95% CI)	*z*	*p*	OR (95% CI)	*z*	*p*	OR (95% CI)	*z*	*p*
Sibling context												
Only-child	1.33 (0.85–2.10)	1.27	0.205	1.71 (1.06–2.75)	2.20	0.028	0.70 (0.43–1.11)	−1.51	0.131	0.69 (0.42–1.13)	−1.47	0.141
Mixed-sex sibling	1.75 (1.07–2.90)	2.22	0.026	1.74 (1.03–2.94)	2.08	0.037	0.80 (0.48–1.34)	−0.85	0.394	0.85 (0.49–1.47)	−0.57	0.568
Same-sex sibling	1			1			1			1		
Sex												
Girls	0.53 (0.40–0.70)	−4.48	<0.001	0.19 (0.14–0.26)	−10.80	<0.001	0.55 (0.42–0.73)	−4.23	<0.001	0.17 (0.13–0.24)	−11.33	<0.001
Boys	1			1			1			1		
Parents’ highest educational level												
Enter/complete college	0.83 (0.63–1.10)	−1.31	0.191	0.79 (0.59–1.05)	−1.61	0.107	1.00 (0.76–1.32)	0	0.999	1.01 (0.76–1.35)	0.08	0.938
High school or below	1			1			1			1		
Mother’s work status												
Working currently	0.80 (0.52–1.22)	−1.03	0.304	0.69 (0.44–1.07)	−1.67	0.095	1.36 (0.86–2.15)	1.33	0.183	0.96 (0.59–1.55)	−0.18	0.860
Not working	1			1			1			1		
Born in Shanghai												
Yes	1.05 (0.70–1.57)	0.25	0.801	0.83 (0.55–1.25)	−0.91	0.365	0.82 (0.56–1.20)	−1.03	0.305	1.20 (0.81–1.78)	0.91	0.362
No	1			1			1			1		
Parental marital status												
Married	0.91 (0.57–1.45)	−0.41	0.683	0.98 (0.60–1.59)	−0.10	0.924	1.04 (0.68–1.59)	0.20	0.845	0.93 (0.59–1.47)	−0.30	0.766
Divorced/separated	1			1			1			1		

**Table 5 children-09-01281-t005:** Ordinal logistic regression testing sibling context on two gender norm scales’ scores, stratified by sex.

	Male						Female					
	GST			GSR			GST			GSR		
	OR (95% CI)	*z*	*p*	OR (95% CI)	*z*	*p*	OR (95% CI)	*z*	*p*	OR (95% CI)	*z*	*p*
Sibling context												
Only-child	0.67 (0.48–0.94)	−2.35	0.019	0.89 (0.64–1.26)	−0.63	0.530	0.99 (0.71–1.39)	−0.06	0.953	0.87 (0.61–1.24)	−0.79	0.429
Same-sex sibling	0.59 (0.37–0.96)	−2.14	0.033	0.89 (0.55–1.45)	−0.46	0.644	1.23 (0.72–2.10)	0.76	0.449	0.72 (0.40–1.31)	−1.09	0.278
Mixed-sex sibling	1			1			1			1		
Age												
13–14	0.99 (0.76–1.30)	−0.06	0.956	1.19 (0.90–1.56)	1.22	0.223	1.04 (0.79–1.36)	0.26	0.796	0.99 (0.74–1.32)	−0.08	0.934
10–12	1			1			1			1		
Parents’ highest educational level												
Enter/complete college	0.93 (0.71–1.22)	−0.53	0.594	1.00 (0.76–1.32)	−0.01	0.989	0.89 (0.68–1.17)	−0.82	0.412	0.79 (0.59–1.06)	−1.55	0.120
High school or below	1			1			1			1		
Mother’s work status												
Working currently	1.27 (0.80–2.01)	1.02	0.308	0.99 (0.61–1.60)	−0.05	0.964	0.86 (0.57–1.31)	−0.69	0.493	0.68 (0.44–1.04)	−1.77	0.076
Not working	1			1			1			1		
Born in Shanghai												
Yes	0.90 (0.62–1.31)	−0.54	0.586	0.90 (0.61–1.31)	−0.57	0.568	0.94 (0.62–1.42)	−0.30	0.763	1.19 (0.77–1.84)	0.79	0.430
No	1			1			1			1		
Parental marital status												
Married	0.88 (0.54–1.42)	−0.53	0.598	0.84 (0.52–1.36)	−0.72	0.473	1.02 (0.68–1.54)	0.10	0.920	1.05 (0.67–1.66)	0.22	0.827
Divorced/separated	1			1			1			1		

## Data Availability

The data presented in this study are available from the corresponding authors on reasonable request. The study used Stata SE 15.1 on the Microsoft Windows 10 operating system to perform analysis. The source code used to generate the results displayed in the tables is also available from the corresponding authors on reasonable request.

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
