# Peer review of "Comparing the Perceptions of Gender Norms among Adolescents with Different Sibling Contexts in Shanghai, China"

_children, 2022, doi:10.3390/children9091281_

Round 1
Reviewer 1 Report
The manuscript presents an interesting article on the relation between the presence of siblings and their sex at birth, with perceived genders' norms and traits in Chinese adolescents. Overall, the paper is clear and coherent, with the discussion following the results.
I have some concerns that i think authors should address before the paper can be considered for publication, mostly regarding the rationale of the study, the methods, and the limitations.
- One of the criticalities for me is the usage of the word adolescence, when it's clear from Line 54 and from the age range considered in the inclusion criteria that what the authors are focusing on is early adolescence. I perceive this as an ethical problem in overselling the results, as well as in a possibly wrong generalization to a wider group. I would suggest the authors to check the whole manuscript for consistency, starting from the title and abstract, and to make sure that the word adolescent is not employed in sentences in which it may appear as an overgeneralization of obtained results.
- Line 128. The section is a bit misleading, as from the first sentence it seems like that the presented analysis is a secondary analysis of data collected for the GEAS survey, while the following lines seems to suggest that the authors collected the data. Given the reference in 32, where data collected in Shangai were limited to the pilot, I feel that the authors recorded the data themselves as part of the GEAS consortium/group, but I may be wrong on this. I would suggest the authors to edit the paragraph to contextualize the data collection in the GEAS research, or to clearly indicate if this is a secondary data analysis.
- Line 94. From the OCP a reader not familiar with the history of China may infer that there are no families with more than one child, and also a limited number of cases in which there is an adolescent child and an older sibling, given that the policy has changed in 2015. As such, I would suggest the authors to provide more details about the OCP and what the numbers of only child are in China now, to provide context for the current research, as well as to indicate, if possible, the number of families with more than one child in China.
- Given the not balanced number of samples for the mother working status, place of birth, and marital status, I would have expected to see a section in the limitations of the study that explain how such a balance could have affected obtained results. I would suggest the authors to add these as possible limitations of the work.
- For reproducibility purposes, please also indicate the platform/operating system in which Stata was used.
Overall the paper is clear and presents an interesting study. Minor spelling check may be required for some missing spaces, typos, and wrong capitalization. I would suggest the editor to accept the paper after some edits to improve the context of this work, clarity and reproducibility are made.
Author Response
Dear reviewer,
Thank you so much for giving us an opportunity to revise our manuscript entitled, “Comparing the perceptions of gender norms among adolescents with different sibling contexts in Shanghai, China”. We would like to thank you for your careful reviews and helpful comments. Our responses to your specific concerns are described below.
Reviewer 1:
- One of the criticalities for me is the usage of the word adolescence, when it's clear from Line 54 and from the age range considered in the inclusion criteria that what the authors are focusing on is early adolescence. I perceive this as an ethical problem in overselling the results, as well as in a possibly wrong generalization to a wider group. I would suggest the authors to check the whole manuscript for consistency, starting from the title and abstract, and to make sure that the word adolescent is not employed in sentences in which it may appear as an overgeneralization of obtained results.
Response: Thank you very much for the comments. Our respondents are adolescents aged 10-14 who were in middle school. Parental consent and youth assent have obtained before we conducted the survey, besides, our survey was conducted in the school environment where teachers were surrounded though they were not present when youth were doing the survey using the pad. And thank you for the reminder. The author has paid attention to the use of adolescence in the revision and changed it to “early adolescence” while changing adolescents to “adolescents aged 10-14” or “early adolescents”.
- Line 128. The section is a bit misleading, as from the first sentence it seems like that the presented analysis is a secondary analysis of data collected for the GEAS survey, while the following lines seems to suggest that the authors collected the data. Given the reference in 32, where data collected in Shangai were limited to the pilot, I feel that the authors recorded the data themselves as part of the GEAS consortium/group, but I may be wrong on this. I would suggest the authors to edit the paragraph to contextualize the data collection in the GEAS research, or to clearly indicate if this is a secondary data analysis.
Response: The data analysis of this study is not secondary data analysis. The data was collected by the authors as part of the GEAS longitudinal survey. The reference you mentioned was the GEAS pilot study which was prior to the GEAS baseline survey. GEAS study groups have agreed that the coordinating site could use their site-specific data without their permission as each site provided the funding for their site to conduct the survey independently in the data sharing memos.
We are sincerely sorry that we cited the wrong reference of the same author we are cooperating with. The one that should be cited here was “Moreau, C., M. Li, S. Ahmed, X. Zuo, and B. Cislaghi. "Assessing the Spectrum of Gender Norms Perceptions in Early Adolescence: A Cross-Cultural Analysis of the Global Early Adolescent Study." J Adolesc Health 69, no. 1s (2021): S16-S22.”. Another paper with more descriptive data about social context written by Mmari et al. was also cited. In these two papers, the description of the data collection in the Shanghai site was consistent with ours.
We revised the description as “Data used in this study were collected in Shanghai from November to December 2017, which is part of the baseline survey of the Global Early Adolescent Study (GEAS), a multi-center longitudinal research focused on gender norms among early adolescents living in disadvantaged urban sites(9,35).” (line 134-137, page 3)
- Line 94. From the OCP a reader not familiar with the history of China may infer that there are no families with more than one child, and also a limited number of cases in which there is an adolescent child and an older sibling, given that the policy has changed in 2015. As such, I would suggest the authors to provide more details about the OCP and what the numbers of only child are in China now, to provide context for the current research, as well as to indicate, if possible, the number of families with more than one child in China.
Response: Thank you very much for pointing this out. We’ve added a more detailed description of the policy as well as the number and proportion of the one child as you suggested as follows (in italics):
The OCP implementation was also influenced by region (e.g., ethnic minorities, or rural families where the first child was a girl in some provinces, were exempted from OCP), parental educational level, family economic level, and other factors. The sixth national census data indicated that until 2015 there were 224 million only children. Of them, 87.3% were born in urban areas. This group would still be significantly large even though the government has encouraged couples to have 2 or 3 children since the one-child policy was abolished in 2015. According to the national family planning survey data, the proportion of only-child dropped from 54.7% in 2006 to 37.0% in 2016. Thus, China would have increasing proportions of adolescents with same- or mixed-sex siblings in the following decades. (Line 95-103, pages 2-3)
- Given the not balanced number of samples for the mother working status, place of birth, and marital status, I would have expected to see a section in the limitations of the study that explain how such a balance could have affected obtained results. I would suggest the authors to add these as possible limitations of the work.
Response: Thank you for the comments.
Balanced data among groups are required in the intervention study but not necessarily required in the cross-sectional study. The unbalanced distributed social demographic factors are probably the influencing factors of the outcome.
However, when the number of observations in certain factor is small, and we attempt to adjust for several factors using methods like logistic regression, it will yield a large standard error and a wide 95% CI, thus we can fail to produce sensible results or produce unreliable results, which might explain that we only got the marginal protective effect (OR:0.68, 95%CI:0.44-1.04, P=0.076) of mothers working status on girls’ higher GSR score.
Besides, given that we only used the cross-sectional data (baseline data of the GEAS Shanghai site), we could not get that the causal relationship between the factors and the outcome. Thus, in the limitation part, we addressed it as follows:
“Lastly, as the number of observations in factors like mother working status, place of birth, and parental marital status is small and we attempt to use logistic regression adjusting for several factors, it can fail to produce sensible results. Besides, given the nature of the cross-sectional data, the result of our study could not guarantee the causal interpretation of the findings as well as generalizability beyond the urban poor early adolescents in this geographical area.”(Line 100-105, page 14)
- For reproducibility purposes, please also indicate the platform/operating system in which Stata was used.
Response: Thank you for the reminder. We’ve revised the description as follows: “The study used Stata SE 15.1 at Microsoft windows 10 operating system to perform analysis.” (Line 133-134, page 15)
Reviewer 2 Report
The topic of the article relates to an interesting research issue, which is the relationship between the perception of gender norms and differentiated sibling contexts. This issue is particularly important in China where, due to the demographic policy of the authorities, a high percentage of young people do not have siblings in their nuclear families. The results obtained by the authors showed that sibling context had a more pronounced influence on gender socialization process for boys than for girls. Moreover, the presence of more traditional gender role attitudes was found in adolescents who are the only children or with same-sex siblings. The authors rightly pointed out some limitations related to the conducted research. However, in the section ‘Conclusions and implications’, there is a lack of specific, more detailed, proposals aimed at people and institutions responsible for the implementation of sex education. I support the publication of the article after taking into account the amendment I proposed
Author Response
Dear reviewer,
Thank you so much for giving us an opportunity to revise our manuscript entitled, “Comparing the perceptions of gender norms among adolescents with different sibling contexts in Shanghai, China”. We would like to thank you for your careful reviews and helpful comments. Our revision to your specific concern regarding the implementation of CSE is described below:
Reviewer 2:
The topic of the article relates to an interesting research issue, which is the relationship between the perception of gender norms and differentiated sibling contexts. This issue is particularly important in China where, due to the demographic policy of the authorities, a high percentage of young people do not have siblings in their nuclear families. The results obtained by the authors showed that sibling context had a more pronounced influence on gender socialization process for boys than for girls. Moreover, the presence of more traditional gender role attitudes was found in adolescents who are the only children or with same-sex siblings. The authors rightly pointed out some limitations related to the conducted research. However, in the section ‘Conclusions and implications’, there is a lack of specific, more detailed, proposals aimed at people and institutions responsible for the implementation of sex education. I support the publication of the article after taking into account the amendment I proposed.
Response: Thank you very much for the comments. We’ve added more detailed suggestions in the conclusion and implication parts as follows:
The World Health Organization pointed out that sexual health is critically influenced by gender norms, roles, expectations, and power dynamics. Yet there is mounting evidence that Comprehensive Sexuality Education (CSE), with the component addressing issues of gender and power, is particularly effective. UNESCO technical guidance regarding CSE could be delivered in formal and non-formal settings, and it is incremental age- and developmentally- appropriate. CSE contributes to gender equality by building awareness of the centrality and diversity of gender in people’s lives; examining gender norms shaped by cultural, social, and biological differences and similarities; and by encouraging the creation of respectful and equitable relationships based on empathy and understanding. Such guidance could be used by parents, school teachers, health workers, and gender professionals to create a more gender-inclusive context for children and adolescents. For parents, sometimes being supportive doesn't mean doing anything more than simply providing a loving and affirmative place for the children to express themselves. For school teachers, building a supportive relationship with respect and equal treatment to all students, and teaching about equitable gender attitudes using lessons, materials, and vignette scenarios would help to their understanding of gender equity. Health workers and gender professionals could offer a professional response to the needs of children and their families and other care providers, to accompany their growth and comprehensive development by respecting their individuality and advocating the full exercise of children’s and adolescents’ gender and power equality. At the societal level, promoting laws and policies to ensure gender equality, organizing media campaigns and large-scale communication programs can contribute to raising awareness. In all, the inequitable gender norms need to be challenged from the macro-level(e.g., promoting laws &policies) to the micro-level (e.g., education both in school and at home) (line 114-138, page 15)